# SPONTANEOUS SYMMETRY BREAKING IN DEEP NEURAL NETWORKS

## ABSTRACT

We propose a framework to understand the unprecedented performance and robustness of deep neural networks using field theory. Correlations between the weights within the same layer can be described by symmetries in that layer, and networks generalize better if such symmetries are broken to reduce the redundancies of the weights. Using a two parameter field theory, we find that the network can break such symmetries itself towards the end of training in a process commonly known in physics as spontaneous symmetry breaking. This corresponds to a network generalizing itself without any user input layers to break the symmetry, but by communication with adjacent layers. In the layer decoupling limit applicable to residual networks (He et al., 2015a), we show that the remnant symmetries that survive the non-linear layers are spontaneously broken based on empirical results. The Lagrangian for the non-linear and weight layers together has striking similarities with the one in quantum field theory of a scalar. Using results from quantum field theory we show that our framework is able to explain many experimentally observed phenomena, such as training on random labels with zero error (Zhang et al., 2017), the information bottleneck and the phase transition out of it and the following increase in gradient variances (Shwartz-Ziv & Tishby, 2017), shattered gradients (Balduzzi et al., 2017), and many more.

## 1 INTRODUCTION

Deep neural networks have been used in image recognition tasks with great success. The first of its kind, AlexNet (Krizhevsky et al., 2012), led to many other neural architectures have been proposed to achieve start-of-the-art results in image processing at the time. Some of the notable architectures include, VGG (Simonyan & Zisserman, 2015), Inception (Szegedy et al., 2015) and Residual networks (ResNet)(He et al., 2015a).

Understanding the inner workings of deep neural networks remains a difficult task. It has been discovered that the training process ceases when it goes through an information bottleneck (Shwartz-Ziv & Tishby, 2017) until learning rate is decreased to a suitable amount then the network under goes a phase transition. Deep networks appear to be able to regularize themselves and able to train on randomly labeled data Zhang et al. (2017) with zero training error. The gradients in deep neural networks behaves as white noise over the layers Balduzzi et al. (2017). And many other unexplained phenomena.

A recent work (Anonymous, 2018) showed that the ensemble behavior and binomial path lengths (Veit et al., 2016) of ResNets can be explained by just a Taylor series expansion to first order in the decoupling limit. They found that the series approximation generates a symmetry breaking layer that reduces the redundancy of weights, leading to a better generalization. Because the ResNet does not contain such symmetry breaking layers in the architecture. They suggest that ResNets are able to break the symmetry by the communication between the layers. Another recent work also employed the Taylor expansion to investigate ResNets (Jastrzebski et al., 2017).

In statistical terms, a quantum theory describes errors from the mean of random variables. We wish to study how error propagate through each layer in the network, layer by layer. In the limit of a continuous sample space, the quantum theory becomes a quantum field theory. The effects of sampling error and labelling error can then be investigated. It is well known in physics that a scalar field can drive a phase transition. Using a scalar field theory we show that a phase transition must

exist towards the end of training based on empirical results. It is also responsible for the remarkable performance of deep networks compared to other classical models. In Appendix D, We explain that quantum field theory is likely one of the simplest model that can describe a deep network layer by layer in the decoupling limit.

Much of the literature on neural network design focuses on different neural architecture that breaks symmetry explicitly, rather than spontaneously. For instance, non-linear layers explicitly breaks the symmetry of affine transformations. There is little discussion on spontaneous symmetry breaking. In neural networks, the Goldstone theorem in field theory states that for every continuous symmetry that is spontaneously broken, there exists a weight with zero Hessian eigenvalue at the loss minimum. No such weights would appear if the symmetries are explicitly broken. It turns out that many seemingly different experimental results can be explained by the presence of these zero eigenvalue weights. In this work, we exploit the layer decoupling limit applicable to ResNets to approximate the loss functions with a power series in symmetry invariant quantities and illustrate that spontaneous symmetry breaking of affine symmetries is the sufficient and necessary condition for a deep network to attain its unprecedented power.

The organization of this paper is as follows. The background on deep neural networks and field theory is given in Section 2. Section 3 shows that remnant symmetries can exist in a neural network and that the weights can be approximated by a scalar field. Experimental results that confirm our theory is given in Section 4. We summarize more evidence from other experiments in Appendix A. A review of field theory is given in Appendix B. An explicit example of spontaneous symmetry breaking is shown in Appendix C.

## 2 BACKGROUND AND FRAMEWORK

In this section we introduce our frame work using a field theory based on Lagrangian mechanics.

### 2.1 DEEP NEURAL NETWORKS

A deep neural network consists of layers of neurons. Suppose that the first layer of neurons with weight matrix $\mathbf{W}_1'$ and bias $\mathbf{b}_1$ takes input $\mathbf{x}_1'$ and outputs $\mathbf{y}_1$

$$\mathbf{y}_1 = \mathbf{W}_1'\mathbf{x}_1' + \mathbf{b}_1 = \mathbf{W}_1\mathbf{x}_1,$$

where $\mathbf{x} = (\mathbf{x}', 1)$ and $\mathbf{W}_1 = (\mathbf{W}_1', \mathbf{b}_1)$, where $\mathbf{W}_1$ and $\mathbf{b}_1$ are real valued. Now suppose that $R_1$ denotes a nonlinear operator corresponding to a sigmoid or ReLU layer located after the weight layer, so that

$$\mathbf{y}_1 = R_1\mathbf{W}_1\mathbf{x}_1.$$

For a neural network with $T$ repeating units, the output for layer $t$ is

$$
\begin{aligned}
\mathbf{y}_t &= R_t\mathbf{W}_t\mathbf{y}_{t-1} \\
&= R_t\mathbf{W}_tR_{t-1}\mathbf{W}_{t-1}\dots R_1\mathbf{W}_1\mathbf{x}_1 \\
&= \left(\prod_{n=0}^{t-1} R_{t-n}\mathbf{W}_{t-n}\right)\mathbf{x}_1.
\end{aligned}
\tag{1}
$$

### 2.2 SYMMETRIES IN NEURAL NETWORKS

We now show the necessary and sufficient conditions of preserving symmetry. We explicitly include symmetry transformations in Equation (1) and investigate the effects caused by a symmetry transformation of the input in subsequent layers. Suppose $Q_t \in G$ is a transformation matrix in some Lie group $G$ for all $t$. Note that the $Q_t$ are not parameters to be estimated. We write $\mathbf{y}_t = \mathbf{y}_t(Q_t)$, where the dependence on $Q_t$ is obtained from a transformation on the input, $\mathbf{x}_t(Q_t) = Q_t\mathbf{x}_t$, and the weights, $\mathbf{W}_t(Q_t)$

$$\mathbf{y}_t(Q_t) = R_t\mathbf{W}_t(Q_t)\mathbf{x}_t(Q_t).$$

If $G$ is a symmetry group, then $\mathbf{y}_t$ is covariant with $\mathbf{x}_t$, such that $\mathbf{y}_t(Q_t) = Q_t\mathbf{y}_t$. This requires two conditions to be satisfied. First, $\mathbf{W}_t(Q_t) = Q_t\mathbf{W}_tQ_t^{-1}$, where $Q_t^{-1}Q_t = I$ and the existence of the inverse is trivial because $G$ is a group and $Q_t \in G$. The second is the commutativity between $R_t$ and

$Q_t$, such that $R_t Q_t = Q_t R_t$. For example, if $g_t \in \text{Aff}(D)$, the group of affine transformations, $R_t$ may not commute with $g_t$. However, commutativity is satisfied when the transformation corresponds to the 2D rotation of feature maps.

Including transformation matrices, the output at layer $t$ is

$$\mathbf{y}_t(Q_t) = \left( \prod_{n=0}^{t-1} R_{t-n} Q_{t-n} \mathbf{W}_{t-n} Q_{t-n}^{-1} \right) Q_1 \mathbf{x}_1.$$

## 2.3 THE LOSS FUNCTIONAL

Statistical learning requires the loss function to be minimized. It can be written in the form of a mutual information, training error, or the Kullback-Leibler divergence. In this section we approximate the loss function in the continuum limit of samples and layers. Then we define the loss functional to transition into Lagrangian mechanics and field theory. Let $\mathbf{z}_i = (\mathbf{X}_i, \mathbf{Y}_i) \in \mathcal{X}$ be the $i$-th input sample in data set $\mathcal{X}$, $(\mathbf{X}_i, \mathbf{Y}_i)$ are the features and the desired outputs, respectively, and $i \in \{1, \ldots, N\}$. The loss function is

$$L = \frac{1}{N} \sum_{i=1}^{N} L_i(\mathbf{X}_i, \mathbf{Y}_i, \mathbf{W}, \mathbf{Q}),$$

where $\mathbf{W} = (\mathbf{W}_1, \ldots, \mathbf{W}_T)$, and $\mathbf{Q} = (Q_1, \ldots, Q_T)$, $Q_t \in G$ where $G$ is a Lie group, and $T$ is the depth of the network. Taking the continuum limit,

$$L \simeq \int_{\mathcal{X}} p(\mathbf{X}, \mathbf{Y}) L_{\mathbf{x}}(\mathbf{X}, \mathbf{Y}, \mathbf{W}, \mathbf{Q}) d\mathbf{X} d\mathbf{Y},$$

where $p(\mathbf{X}, \mathbf{Y})$ is the joint distribution of $\mathbf{X}$ and $\mathbf{Y}$. Using the first fundamental theorem of calculus and taking the continuous layers ($t$) limit, we write

$$L_{\mathbf{x}} \simeq L_{\mathbf{x}}(t = 0) + \int_0^T \frac{dL_{\mathbf{x}}(\mathbf{X}, \mathbf{Y}, \mathbf{W}(t), Q(t))}{dt} dt,$$

where $L_{\mathbf{x}}(t = 0)$ is the value of the loss before training. We let $L_{\mathbf{x},t} = dL_{\mathbf{x}}/dt$ be the loss rate per layer. The loss rate $L_{\mathbf{x},t}$ is bounded from below. Therefore

$$\min_{\mathbf{W}} L_{\mathbf{x}}(\mathbf{X}, \mathbf{Y}, \mathbf{W}, \mathbf{Q}) = L_{\mathbf{x}}(t = 0) + \int_0^T \min_{\mathbf{W}} L_{\mathbf{x},t}(\mathbf{X}, \mathbf{Y}, \mathbf{W}(t), Q(t)) dt.$$

Minimizing the loss rate guarantees the minimization of the total loss. We require $L_{\mathbf{x},t}$ to be invariant under symmetry transformations. That is, if $Q_1(t), Q_2(t) \in G$. Then

$$L_{\mathbf{x},t}(\mathbf{X}, \mathbf{Y}, \mathbf{W}(t), Q_1(t)) = L_{\mathbf{x},t}(\mathbf{X}, \mathbf{Y}, \mathbf{W}(t), Q_2(t)).$$

However if $Q_1(t)$ and $Q_2(t)$ do not belong in the same symmetry group, the above equality does not necessarily hold. Now we define the loss functional for a deep neural network

$$\mathcal{P}[\mathbf{W}, Q] = \int_{\mathcal{X} \times \{0, T\}} p(\mathbf{X}, \mathbf{Y}) L_{\mathbf{x},t}(\mathbf{X}, \mathbf{Y}, \mathbf{W}(t), Q(t)) d\mathbf{X} d\mathbf{Y} dt.$$

## 2.4 LAGRANGIAN MECHANICS AND FIELD THEORY

Having defined the loss functional, we can transition into Lagrangian dynamics to give a description of the feature map flow at each layer. Let the minimizer of the loss rate be

$$\mathbf{W}^*(\mathbf{X}, \mathbf{Y}, Q(t), t) = \arg\min_{\mathbf{W}} p(\mathbf{X}, \mathbf{Y}) L_{\mathbf{x},t}(\mathbf{X}, \mathbf{Y}, \mathbf{W}(t), Q(t)).$$

From now on, we combine $\mathbf{z} = (\mathbf{X}, \mathbf{Y})$ as $\mathbf{Y}$ only appears in $\mathbf{W}^*$ in this formalism, each $\mathbf{Y}$ determines a trajectory for the representation flow determined by Lagrangian mechanics. Now we define, for each $i$-th element of $\mathbf{W}(t)$, and a non-linear operator $R(t)$ acting on $\mathbf{W}(t)$ such that the loss minimum is centered at the origin,

$$w^i(\mathbf{z}, Q(t), t) = R(t) W^i(\mathbf{z}, Q(t), t) - R(t) W^{*i}(\mathbf{z}, Q(t), t). \tag{2}$$

We now define the Lagrangian density,

$$\mathcal{L} = \mathcal{L}(\mathbf{z}, \mathbf{w}, \partial_t \mathbf{w}, \partial_\mathbf{z} \mathbf{w}, Q(t))$$

and $\mathcal{L} = \mathcal{T} - \mathcal{V}$, where $\mathcal{T}$ is the kinetic energy and $\mathcal{V}$ is the potential energy. We define the potential energy to be

$$\mathcal{V} = p(\mathbf{z}) L_{\mathbf{x},t}(\mathbf{z}, \mathbf{W}(t), Q(t)),$$

The probability density $p(\mathbf{z})$ and the loss rate $L_{\mathbf{x},t}$ are invariant under symmetry transformations. Therefore $\mathcal{V}$ is an invariant as well.

**Definition: Orthogonal Group**   The orthogonal group O($D$) is the group of all $D \times D$ matrices such that $O^T O = I$.

We now set up the conditions to obtain a series expansion of $\mathcal{V}$ around the minimum $w^i(Q_t) = 0$. First, since $\mathcal{V}$ is an invariant. Each term in the series expansion must be an invariant such that $f(w_i(Q_t), w^i(Q_t)) = f(w_i, w^i)$ for all $Q_t \in G$. Suppose $G = O(D)$, the orthogonal group and that $\mathbf{w}^T(Q_t) = \mathbf{w}^T Q^T$ and $\mathbf{w}(Q_t) = Q_t \mathbf{w}$. So $w_i w^i$ is an invariant. Then $f = w_i w^i$ is invariant for all $Q_t$ where the Einstein summation convention was used

$$w_i w^i = \sum_{i=1}^{D} w_i w^i = \mathbf{w}^T \mathbf{w},$$

Now we perform a Taylor series expansion about the minimum $w^i = 0$ of the potential,

$$\mathcal{V} = C + w_i H^i_j w^j + w_i w_j \Lambda^{ij}_{mn} w^m w^n + O((w_i w^i)^6),$$

where $H^i_j = \partial_{w^i} \partial_{w^j} \mathcal{V}$ is the Hessian matrix, and similarly for $\Lambda^{ij}_{mn}$. The overall constant $C$ can be ignored without loss of generality. Because $\mathcal{V}$ is an even function in $w^i$ around the minimum, we must have

$$w_i H^i_j w^j = \sum_{i=1}^{D} w_i H^i_i w^i.$$

The O($D$) symmetry enforces that all weight Hessian eigenvalues to be $H^i_i = m^2/2$ for some constant $m^2$. This can be seen in the O(2) case, with constants $a, b, a \neq b, Q \in$ O(2) such that $w_1(Q) = w_2$ and $w_2(Q) = w_1$,

$$a w_1(Q)^2 + b w_2(Q)^2 = a w_2^2 + b w_1^2,$$

this does not equal $a w_1^2 + b w_2^2$, so the O(2) symmetry implies $a = b$. This can be generalized to the O($D$) case. For the quartic term, the requirement that $\mathcal{V}$ be even around the minimum gives

$$w_i w_j \Lambda^{ij}_{mn} w^m w^n = \sum_{i,j=1}^{D} \Lambda^{ij}_{ij} w_j w^j w_i w^i.$$

Similarly the O($D$) symmetry implies $\Lambda^{ii}_{ii} = \lambda/4$ for some constant $\lambda$ and zero for any other elements, the potential is

$$\mathcal{V} = \frac{m^2}{2} w_i w^i + \frac{\lambda}{4} (w_i w^i)^2,$$

where the numerical factors were added for convention. The power series is a good approximation in the decoupling limit which may be applicable for Residual Networks.[1] For the kinetic term $\mathcal{T}$, we expand in power series of the derivatives,

$$\mathcal{T} = \frac{1}{2} \frac{\partial w_i}{\partial t} \frac{\partial w^i}{\partial t} - \frac{c^2}{2} \frac{\partial w_i}{\partial \mathbf{z}} \frac{\partial w^i}{\partial \mathbf{z}} + O((\partial_t w)^4, (\partial_\mathbf{z} w)^4),$$

---

[1]The output of a residual unit is $\mathbf{y}_t = \mathbf{x}_t + \mathbf{F}_t(\mathbf{x}_t, \mathbf{W}_{1t}, \mathbf{W}_{2t})$, with $\mathbf{F}_t \ll \mathbf{x}_t$ and $\mathbf{F}_t(\mathbf{x}_t, \mathbf{W}_{1t}, \mathbf{W}_{2t}) = R_{t2} \mathbf{W}_{t2} R_{t1} \mathbf{W}_{t1} \mathbf{x}_t$. After centering the loss minimum at the origin with Equation 2 and suitably normalizing, we can rewrite this as $\mathbf{F}_t = a \mathbf{w}^T \mathbf{w}$, with $a \ll 1$ in the decoupling limit. See Peskin & Schroeder (1995) for the normalization of a scalar field.

where the coefficient for $(\partial_t w)^2$ is fixed by the Hamiltonian kinetic energy $\frac{1}{2}(\partial_t w)^2$. Higher order terms in $(\partial_t w)^2$ are negligible in the decoupling limit. If the model is robust, then higher order terms in $(\partial_\mathbf{z} w)^2$ can be neglected as well. [2] The Lagrangian density is [3]

$$\mathcal{L} = \frac{1}{2}(\partial_t w)^2 - \frac{1}{2}(\partial_\mathbf{z} w)^2 - \frac{m^2}{2}w^2 - \frac{\lambda}{4}(w^2)^2,$$

where we have set $w^2 = w_i w^i$ and absorbed $c$ into $\mathbf{z}$ without loss of generality. This is precisely the Lagrangian for a scalar field in field theory. Standard results for a scalar field theory can be found in Appendix B. To account for the effect of the learning rate, we employ results from thermal field theory (Kapusta & Gale, 2006) and we identify the temperature with the learning rate $\eta$. So that now $m^2 = -\mu^2 + \frac{1}{4}\lambda\eta^2$, with $\mu^2 > 0$.

## 2.5 SPONTANEOUS SYMMETRY BREAKING

Spontaneous symmetry breaking describes a phase transition of a deep neural network. Consider the following scalar field potential invariant under $O(D')$ transformations,

$$\mathcal{V}(w^i, \eta) = \frac{m^2(\eta)}{2}w_i w^i + \frac{\lambda}{4}(w_i w^i)^2,$$

where $m^2(\eta) = -\mu^2 + \frac{1}{4}\lambda\eta^2$, $\mu^2 > 0$ and learning rate $\eta$. There exists a value of $\eta = \eta_c$ such that $m^2 = 0$. In the first phase, $\eta > \eta_c$, the loss minimum is at $w_{0i}^* = 0$, where

$$w_{0i}^*(\eta) = \arg\min_{w^i} \mathcal{V}(w^i, \eta).$$

When the learning rate $\eta$ drops sufficiently low, the symmetry is spontaneously broken and the phase transition begins. The loss minimum bifurcates at $\eta = \eta_c$ into

$$w_i^*(\eta < \eta_c) = \pm\sqrt{\frac{-m^2(\eta)}{\lambda}}.$$

This occurs when the Hessian eigenvalue becomes negative, $m^2(\eta) < 0$, when $\eta < \eta_c$.

This phenomenon has profound implications. It is responsible for phase transition in neural networks and generates long range correlation between representations and the desired output. Details from field theory can be found in Appendix C. Figure 1 depicts the shape of the loss rate during spontaneous symmetry breaking with a single weight $w$, and the orthogonal group $O(D')$ is reduced to a reflection symmetry $O(1) = \{1, -1\}$ such that $w(Q) = \pm w$. At $\eta > \eta_c$, the loss rate has a loss minima at point $A$. When the learning rate decreases, such that $\eta < \eta_c$, the critical point at $A$ becomes unstable and new minima with equal loss rate are generated. The weight must go through $B$ to get to the new minimum $C$. If the learning rate is too small, the weight will be stuck near $A$. This explains why a cyclical learning rate can outperform a monotonic decreasing learning rate (Smith & Topin, 2017).

Because the loss rate is invariant still to the spontaneously *broken* symmetry, any new minima generated from spontaneous symmetry breaking must have the same loss rate. If there is a unbroken continuous symmetry remaining, there would be a connected loss rate surface corresponding to the new minima generated by the unbroken symmetry. Spontaneous symmetry breaking splits the weights into two sets, $w \to (\pi, \sigma)$. The direction along this degenerate minima in weight space corresponds to $\pi$. And the direction in weight space orthogonal to $\pi$ is $\sigma$. This has been shown experimentally by Goodfellow et al. (2014) in Figure 19. We show the case for the breaking of $O(3)$ to $O(2)$ in Figure 2.

---

[2] The kinetic term $\mathcal{T}$ is not invariant under transformation $Q(t)$. To obtain invariance $\partial_t w^i$ is to be replaced by the covariant derivative $D_t w^i$ so that $(D_t w^i)^2$ is invariant under $Q(t)$(Peskin & Schroeder, 1995). The covariant derivative is $D_t w^i(\mathbf{z}, t, Q(t)) = \partial_t w^i(\mathbf{z}, t, Q(t)) + \alpha B(\mathbf{z}, t, Q(t))_j^i w^j(\mathbf{z}, t, Q(t))$, with $\mathbf{B}(\mathbf{z}, t, Q_t) = Q(t)B(\mathbf{z}, t)Q(t)^{-1}$. The new fields $\mathbf{B}$ introduced for invariance is not responsible for spontaneous symmetry breaking, the focus of this paper. So we will not consider them further.

[3] Formally, the $\partial_\mathbf{z} w$ term should be part of the potential $\mathcal{V}$, as $\mathcal{T}$ contains only $\partial_t w$ terms. However we adhere to the field theory literature and put the $\partial_\mathbf{z} w$ term in $\mathcal{T}$ with a minus sign.

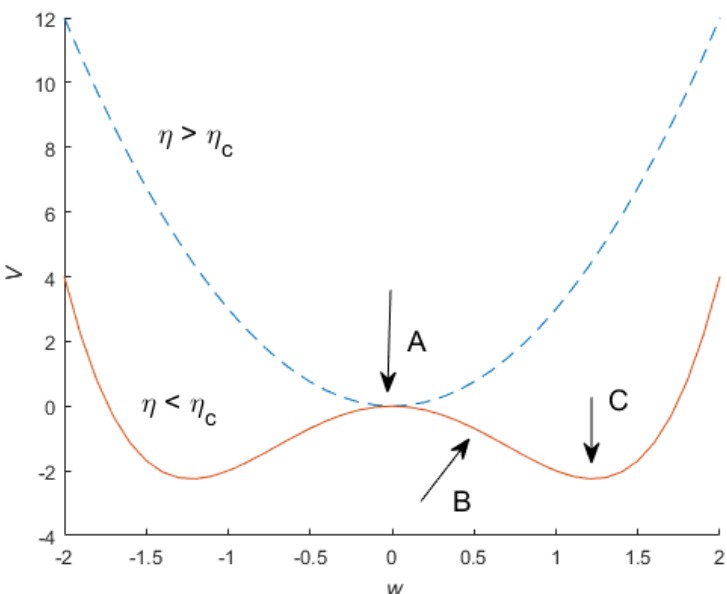

Figure 1: The characteristics of the loss rate with spontaneous symmetry breaking. The dashed line corresponds to the symmetric phase, while the solid line corresponds to the broken phase. Only at $w = 0$ the reflection symmetry $w(Q) = \pm w$ is respected. Here, $w = \sigma$.

## 3 SYMMETRIES IN NEURAL NETWORKS

In this section we show that spontaneous symmetry breaking occurs in neural networks. First, we show that learning by deep neural networks can be considered solely as breaking the symmetries in the weights. Then we show that some non-linear layers can preserve symmetries across the non-linear layers. Then we show that weight pairs in adjacent layers, but not within the same layer, is approximately an invariant under the remnant symmetry leftover by the non-linearities. We assume that the weights are scalar fields invariant under the affine Aff($D'$) group for some $D'$ and find that experimental results show that deep neural networks undergo spontaneous symmetry breaking.

**Theorem 1: Deep feedforward networks learn by breaking symmetries**  *Proof*: Let $A_i$ be an operator representing any sequence of layers, and let a network formed by applying $A_i$ repeatedly such that $x_{out} = (\prod_{i=1}^{M} A_i)x_{in}$. Suppose that $A_i \in$ Aff($D$), the symmetry group of all affine transformations. We have $L = \prod_{i=1}^{D} A_i$, where $L \in$ Aff($D$). Then $x_{out} = Lx_{in}$ for some $L \in$ Aff($D$) and $x_{out}$ can be computed by a single affine transformation $L$. When $A_i$ contains a non-linearity for some $i$, this symmetry is explicitly broken by the nonlinearity and the layers learn a more generalized representation of the input. $\square$

Now we show that ReLU preserves some continuous symmetries.

**Theorem 2: ReLU reduces the symmetry of an Aff($D$) invariant to some subgroup Aff($D'$), where $D' < D$.**  *Proof*: Suppose $R$ denotes the ReLU operator with output $\mathbf{y}_t$ and $Q_t \in$ Aff($D$) acts on the input $\mathbf{x}_t$, where $R(\mathbf{x}) = \max(0, \mathbf{x})$. Let $\mathbf{x}^T\mathbf{x}$ be an invariant under Aff($D$) and let $\mathbf{x}^T = (\gamma, \nu)$, $\nu < 0$ and $\gamma > 0$. Let $\mathbf{a} = R\mathbf{x} = (\gamma, 0)$. Then $\mathbf{a}^T\mathbf{a} = \mathbf{x}^T RR\mathbf{x} = \gamma^T\gamma$. Then $\mathbf{a}^T\mathbf{a}$ is an invariant under Aff($D'$) where $D' = \dim\gamma$. Note that $\gamma^i$ can be transformed into a negative value as it has passed the ReLU already. $\square$

**Corollary**  If there exists a group $G$ that commutes with a nonlinear operator $R$, such that $QR = RQ$, for all $Q \in G$, then $R$ preserves the symmetry $G$.

**Definition: Remnant Symmetry** If $Q_t \in G$ commutes with a non-linear operator $R_t$ for all $Q_t$, then $G$ is a remnant symmetry at layer $t$.

For the loss function $L_i(\mathbf{X}_i, \mathbf{Y}_i, \mathbf{W}, \mathbf{Q})$ to be invariant, we need the predicted output $\mathbf{y}_T$ to be covariant with $\mathbf{x}_i$. Similarly for an invariant loss rate $L_{\mathbf{x},t}$ we require $\mathbf{y}_t$ to be covariant with $\mathbf{x}_t$. The following theorem shows that a pair of weights in adjacent layers can be considered an invariant for power series expansion.

**Theorem 3: Neural network weights in adjacent layers form an approximate invariant** Suppose a neural network consists of affine layers followed by a continuous non-linearity, $R_t$, and that the weights at layer $t$, $\mathbf{W}_t(Q_t) = Q_t \mathbf{W}_t Q_t^{-1}$, and that $Q_t \in H$ is a remnant symmetry such that $Q_t R_t = R_t Q_t$. Then $\mathbf{w}_t \mathbf{w}_{t-1}$ can be considered as an invariant for the loss rate.

*Proof:* Consider $\mathbf{x}(Q_t) = Q_t \mathbf{x}_t$, then

$$
\begin{aligned}
\mathbf{y}_t(Q_t) &= R_t \mathbf{W}_t(Q_t) \mathbf{x}_t(Q_t) \\
&= R_t Q_t \mathbf{W}_t Q_t^{-1} Q_t \mathbf{x}_t \\
&= R_t Q_t \mathbf{W}_t \mathbf{x}_t \\
&= Q_t R_t \mathbf{W}_t \mathbf{x}_t,
\end{aligned}
$$

where in the last line $Q_t R_t = R_t Q_t$ was used, so $\mathbf{y}_t(Q_t) = Q_t \mathbf{y}_t$ is covariant with $\mathbf{x}_t$. Now, $\mathbf{x}_t = R_{t-1} \mathbf{W}_{t-1} \mathbf{x}_{t-1}$, so that

$$
\mathbf{y}_t(Q_t) = Q_t R_t \mathbf{W}_t R_{t-1} \mathbf{W}_{t-1} \mathbf{x}_{t-1}.
$$

The pair $(R_t \mathbf{W}_t)(R_{t-1} \mathbf{W}_{t-1})$ can be considered an invariant under the ramnant symmetry at layer $t$. Let $\mathbf{w}_t = R_t \mathbf{W}_t - R_t \mathbf{W}_t^*$. Therefore $\mathbf{w}_t \mathbf{w}_{t-1}$ is an invariant. $\square$

In the continuous layer limit, $\mathbf{w}_t \mathbf{w}_{t-1}$ tends to $\mathbf{w}(t)^T \mathbf{w}(t)$ such that $\mathbf{w}(t)$ is the first layer and $\mathbf{w}(t)^T$ corresponds to the one after. Therefore $\mathbf{w}(t)$ can be considered as $D'$ scalar fields under the remnant symmetry. The remnant symmetry is not exact in general. For sigmoid functions it is only an approximation. The crucial feature for the remnant symmetry is that it is continuous so that strong correlation between inputs and outputs can be generated from spontaneous symmetry breaking. In the following we will only consider exact remnant symmetries. We will state the Goldstone Theorem from field theory without proof.

**Theorem (Goldstone)** For every spontaneously broken continuous symmetry, there exist a weight $\pi$ with zero eigenvalue in the Hessian $m_\pi^2 = 0$. $\square$

In any case, we will adhere to the case where the remnant symmetry is an orthogonal group $O(D')$. Note that $\mathbf{W}$ is a $D \times D$ matrix and $D' < D$. We choose a subset $\mathbf{\Gamma} \in \mathbb{R}^{D'}$ of $\mathbf{W}$ such that $\mathbf{\Gamma}^T \mathbf{\Gamma}$ is invariant under $\text{Aff}(D')$. Now that we have an invariant, we can write down the Lagrangian for a deep feedforward network for the weights responsible for spontaneous symmetry breaking.

**The Lagrangian for deep feedforward networks in the decoupling limit** Let $\gamma^i = R(\Gamma^i) - R(\Gamma^{*i})$,

$$
\mathcal{L} = \frac{1}{2}(\partial_t \gamma^i)^2 - \frac{1}{2}(\partial_{\mathbf{z}} \gamma^i)^2 - \frac{m^2(\eta)}{2} \gamma^i \gamma_i - \frac{\lambda}{4}(\gamma^i \gamma_i)^2. \tag{3}
$$

Now we can use standard field theory results and apply it to deep neural networks. A review for field theory is given in Appendix B. The formalism for spontaneous symmetry breaking is given in Appendix C.

## 4 MAIN RESULTS

In this section we assume that the non-linear operator is a piecewise linear function such as ReLU and set $R = I$ to be the identity and restrict our attention to the symmetry preserving part of $R$ (see theorem 2). Our discussion also applies to other piecewise-linear activation functions. According to the Goldstone theorem, spontaneous symmetry breaking splits the set of weight deviations $\gamma$ into two sets $(\sigma, \pi)$ with different behaviors. Weights $\pi$ with zero eigenvalues and a spectrum dominated

by small frequencies $\mathbf{k}$ in its correlation function.[4] The other weights $\sigma$, have Hessian eigenvalues $\mu^2$ as the weights before the symmetry is broken. In Appendix C, a standard calculation in field theory shows that the correlation functions of the weights have the form

$$P_{\sigma,\pi}(t, \mathbf{k}) = \frac{i}{2\omega_0} \exp\left(-i\omega_0 t\right), \tag{4}$$

where $\omega_0 = \sqrt{|\mathbf{k}|^2 + m_{\sigma,\pi}^2}$, $m_\sigma^2 = -\mu_\sigma^2 + \frac{1}{4}\lambda\eta^2$ and $m_\pi^2 = \frac{1}{4}\lambda\eta^2$. The correlation function of $\pi$ approaches infinity as $\mathbf{k} \to 0$, as $m_\pi^2 \to 0$. As the loss is minimized, this corresponds to large correlation between representations and the desired output over all sample space *and* layers $t$.

**Spontaneous symmetry breaking and the information bottleneck**     The neural network undergoes a phase transition out of the information bottleneck via spontaneous symmetry breaking described in Section 2.5. Before the phase transition, the weights $\gamma$ have positive Hessian eigenvalues $m^2$. After the phase transition, weights $\pi$ with zero Hessian eigenvalues are generated by spontaneous symmetry breaking. The correlation function for the $\pi$ weights is concentrated around small values of $|\mathbf{k}|$, see Equation (4), with $\omega_0 = |\mathbf{k}|$ for any $t$. This corresponds to a highly correlated representations across the sample (input) space and layers. Because the loss is minimized, the feature maps across the network is highly correlated with the desired output. And a large correlation across the sample space means that the representations are independent of the input. This is shown in Figure 2 of Shwartz-Ziv & Tishby (2017). After phase transition, $I(Y;T) \simeq 1$ bit for all layers $T$, and $I(X;T)$ is small even for representations in early layers.

**Gradient variance explosion**     It has been shown that the variance in weight gradients in the same layer grow by an order of magnitude during the end of training (Shwartz-Ziv & Tishby, 2017). We also connect this to spontaneous symmetry breaking. As two sets of weights, $(\sigma, \pi)$ are generated with different distributions. Considering them as the same object would result in a larger variance.

**Robustness of deep neural networks**     We find that neural networks are resilient to overfitting. Recall that the fluctuation in the weights can arise from sampling noise. Then $(\partial_{\mathbf{z}}w^i)^2$ can be a measure of model robustness. A small value denotes the weights' resistance to sampling noise. If the network were to overfit, the weights would be very sensitive to sampling error. After spontaneous symmetry breaking, weights at the loss minimum with zero eigenvalues obey the Klein-Gordon equation with $m_\pi^2 = 0$,

$$(\partial_{\mathbf{z}}\pi)^2 = (\partial_{\mathbf{z}}\pi^*)(\partial_{\mathbf{z}}\pi) = |\mathbf{k}|^2, \quad \pi = \exp(i\omega t - i\mathbf{k} \cdot \mathbf{z}).$$

The singularity in the correlation function suggests $|\mathbf{k}|^2 \simeq 0$. The zero eigenvalue weights provide robustness to the model. Zhang et al. (2017) referred to this phenomenon as implicit regularization.

## 5   CONCLUDING REMARKS

In this work we solved one of the most puzzling mysteries of deep learning by showing that deep neural networks undergo spontaneous symmetry breaking. This is a first attempt to describe a neural network with a scalar quantum field theory. We have shed light on many unexplained phenomenon observed in experiments, summarized in Appendix A.

One may wonder why our theoretical model works so well explaining the experimental results with just two parameters. It is due to the decoupling limit such that a power series in the loss function is a good approximation to the network. In our case, the two expansion coefficients are the lowest number of possible parameters that is able to describe the phase transition observed near the end of training, where the performance of the deep network improves drastically. It is no coincidence that our model can explain the empirical observations after the phase transition. In fact, our model can describe, at least qualitatively, the behaviors of phase transition in networks that the decoupling limit may not apply to. This suggests that the interactions with nearby layers are responsible for the phase transition.

---

[4]The correlation functions $\langle \pi_i(\mathbf{z}, \mathbf{t})\pi_i(\mathbf{z}', t') \rangle$ for $i \in \{1, \ldots, D'\}$, is a measure of similarity of the weight across at $\mathbf{z}$ and $\mathbf{z}'$. A singularity in frequency domain at $|\mathbf{k}| = 0$ corresponds to a high correlation between the weights across all sample space $\mathbf{z} = (\mathbf{X}, \mathbf{Y})$ and layers $t$. This can be interpreted as the correlation length.

## A  VALIDATION OF PROPOSED FRAMEWORK FOR NEURAL NETWORKS IN THE LITERATURE

In this section we summarize other experimental findings that can be explained by the proposed field theory and the perspective of symmetry breaking. Here $Q \in G$ acts on the the input and hidden variables $\mathbf{x}, \mathbf{h}$, as $Q\mathbf{x}, Q\mathbf{h}$.

- The shape of the loss function after spontaneous symmetry breaking has the same shape observed by Goodfellow et al. (2014) towards the end of training, see Figure 1.

- The training error typically drops drastically when learning rate is decreased. This occurs when the learning rate drops below $\eta_c$, forcing a phase transition so that new minima develop. See Figure 1.

- A cyclical learning rate (Smith & Topin, 2017) helps to get to the new minimum faster, see Section 2.5.

- Stochasticity in gradient descent juggles the loss function such that the weights are no longer at the local maximum of Figure 1. A gradient descent step is taken to further take the weights towards the local minimum. Stochasticity helps the network to generalize better.

- When the learning rate is too small to move away from $A$ in Figure 1. PReLU's (He et al., 2015b) could move the weight away from $A$ through the training of the non-linearity. This corresponds to breaking the symmetry explicitly in Theorem 1.

- Results from Shwartz-Ziv & Tishby (2017) are due to spontaneous symmetry breaking, see Section 4.

- Deep neural networks can train on random labels with low training loss as feature maps are highly correlated with their respective desired output. Zhang et al. (2017) observed that a deep neural network can achieve zero training error on random labels. This shows that small Hessian eigenvalues is not the only condition that determines robustness.

- Identity mapping outperforms other skip connections (He et al., 2016) is a result of the residual unit's output being small. Then the residual units can be decoupled leading to a small $\lambda$ and so it is easier for spontaneous symmetry breaking to occur, from $m^2 = -\mu^2 + \frac{1}{4}\lambda\eta^2$.

- Skip connection across residual units breaks additional symmetry. Suppose now an identity skip connection connects $\mathbf{x}_1$ and the output of $\mathbf{F}_2$. Now perform a symmetry transformation on $\mathbf{x}_1$ and $\mathbf{x}_2$, $Q_1$ and $Q_2 \in G$, respectively. Then the output after two residual untis is $Q\mathbf{x_3} = Q_1\mathbf{x_1} + Q_2\mathbf{x}_2 + Q_2\mathbf{F}_2$. Neither $Q = Q_1$ nor $Q = Q_2$ can satisfy the covariance under $G$. This is observed by Orhan & Pitkow (2017).

- The shattered gradient problem (Balduzzi et al., 2017). It is observed that the gradient in deep (non-residual) networks is very close to white noise. This is reflected in the exponential in Equation (7). This effect on ResNet is reduced because of the decoupling limit $\lambda \to 0$. This leads to the weight eigenvalues $m^2$ being larger in non-residual networks owing to $m^2 = -\mu^2 + \frac{1}{4}\lambda\eta^2$. And so a higher oscillation frequency in the correlation function.

- In recurrent neural networks, multiplicative gating (Yuhuai et al., 2016) combines the input $\mathbf{x}$ and the hidden state $\mathbf{h}$ by an element-wise product. Their method outperforms the method with an addition $\mathbf{x}+\mathbf{h}$ because the multiplication gating breaks the covariance of the output. A transformation $Q\mathbf{x} * Q\mathbf{h} \neq Q(\mathbf{x} * \mathbf{h})$, whereas for addition the output remains covariant $Q\mathbf{x} + Q\mathbf{h} = Q(\mathbf{x} + \mathbf{h})$.

## B  REVIEW OF FIELD THEORY

In this section we state the relevant results in field theory without proof. We use Lagrangian mechanics for fields $w(\mathbf{x}, t)$. Equations of motion for fields are the solution to the Euler-Lagrange equation, which is a result from the principle of least action. The action, $S$, is

$$S[w] = \int L(t, w, \partial_t w)dt,$$

where $L$ is the Lagrangian. Define the Lagrangian density

$$L(t) = \int \mathcal{L}(\mathbf{z}, t, w, \partial_{\mathbf{z},t} w) d\mathbf{z}.$$

The action in term of the Lagrangian density is

$$S[w] = \int \mathcal{L}(\mathbf{z}, t, w, \partial_{\mathbf{z},t} w) d\mathbf{z} dt.$$

The Lagrangian can be written as a kinetic term $\mathcal{T}$, and a potential term $\mathcal{V}$ (loss function),

$$\mathcal{L} = \mathcal{T} - \mathcal{V}$$

For a real scalar field $w(\mathbf{x}, t)$,

$$\mathcal{T} = \frac{1}{2}\left(\frac{\partial w}{\partial t}\right)^2 - \frac{1}{2}\left(\frac{\partial w}{\partial \mathbf{z}}\right)^2 = \frac{1}{2}(\partial_t w)^2 - \frac{1}{2}(\partial_{\mathbf{z}} w)^2$$

where we have set the constant $c^2 = 1$ without loss of generality. The potential for a scalar field that allows spontaneous symmetry breaking has the form

$$\mathcal{V} = \frac{m^2}{2} w^2 + \frac{\lambda}{4} w^4.$$

In the decoupling limit, $\lambda \to 0$, the equation of motion for $w$ is the Klein-Gordon Equation

$$[(\partial_t)^2 - (\partial_{\mathbf{z}})^2 - m^2] w = 0.$$

In the limit of $m^2 \to 0$, the Klein-Gordon Equation reduces to the wave equation with solution

$$w(\mathbf{z}, t) = e^{i(\omega t - \mathbf{k} \cdot \mathbf{z})},$$

where $i = \sqrt{-1}$.

One can treat $w$ as a random variable such that the probability distribution (a functional) of the scalar field $w(\mathbf{z}, t)$ is $p[w] = \exp(-S[w])/\mathcal{Z}$, where $\mathcal{Z}$ is some normalizing factor. The distribution peaks at the solution of the Klein-Gordon equation since it minimizes the action $S$. Now we can define the correlation function between $w(\mathbf{z}_1, t_1)$ and $w(\mathbf{z}_2, t_2)$,

$$\langle w(\mathbf{z}_1, t_1) w(\mathbf{z}_2, t_2) \rangle = \frac{1}{\mathcal{Z}} \int w(\mathbf{z}_1, t_1) w(\mathbf{z}_2, t_2) e^{-S[w]} \mathcal{D}w,$$

where $\int \mathcal{D}w$ denotes the integral over all paths from $(\mathbf{z}_1, t_1)$ to $(\mathbf{z}_2, t_2)$. In the decoupling limit $\lambda \to 0$, it can be shown that

$$\exp(-S[w]) = \exp\left(-\int w(\partial_t^2 - \partial_{\mathbf{z}}^2 + m^2) w \, d\mathbf{z} dt\right),$$

where Stokes theorem was used and the term on the boundary of (sample) space is set to zero. The above integral in the exponent is quadratic in $w$ and the integral over $\mathcal{D}w$ can be done in a similar manner to Gaussian integrals. The correlation function of the fields across two points in space and time is

$$\langle w(\mathbf{z}_1, t_1) w(\mathbf{z}_2, t_2) \rangle = G(\mathbf{z}_1, t_1, \mathbf{z}_2, t_2),$$

where $G(\mathbf{z}_1, t_1, \mathbf{z}_2, t_2)$ is the Green's function to the Klein-Gordon equation, satisfying

$$(\partial_t^2 - \partial_{\mathbf{z}}^2 + m^2) G(\mathbf{z}_1, t_1, \mathbf{z}_2, t_2) = \delta(\mathbf{z}_1 - \mathbf{z}_2) \delta(t_1 - t_2).$$

The Fourier transformation of the correlation function is

$$G(\omega, \mathbf{k}) = \frac{1}{\omega^2 - |\mathbf{k}|^2 - m^2}, \quad m^2 > 0.$$

An inverse transform over $\omega$ gives

$$G(t, \mathbf{k}) = \frac{i}{2\omega_0} e^{-i\omega_0 t},$$

with $\omega_0^2 = |\mathbf{k}|^2 + m^2$.

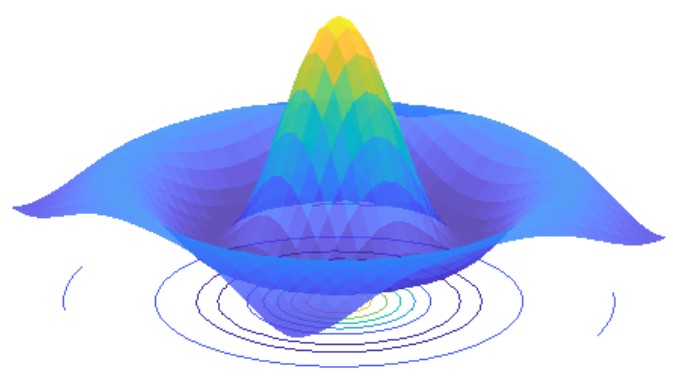

Figure 2: The loss after spontaneous symmetry breaking from othogonal group O(3) to O(2). The symmetry transformations of O(2), which are 1-D rotations, form the degenerate minima.

## C  SPONTANEOUS SYMMETRY BREAKING IN THE ORTHOGONAL GROUP O($D'$)

In this section we show that weights $\pi$ with small, near zero, eigenvalues $m_\pi^2 = \frac{1}{4}\lambda\eta^2$ are generated by spontaneous symmetry breaking. Note that we can write the Lagrangian in Equation (3) as $\mathcal{L} = \mathcal{T} - \mathcal{V}$. Consider weights $\gamma$ that transforms under O($D'$), from Equation (3)

$$
\begin{aligned}
\mathcal{T} &= \frac{1}{2}(\partial_t \gamma^i)^2 - \frac{1}{2}(\partial_\mathbf{z} \gamma^i)^2, \\
\mathcal{V} &= \frac{m^2}{2}\gamma^i \gamma_i + \frac{\lambda}{4}(\gamma^i \gamma_i)^2.
\end{aligned}
\tag{5}
$$

When $m^2 = -\mu^2 + \frac{1}{4}\lambda\eta^2 < 0$, it can be shown that in this case the loss minimum is no longer at $\gamma^i = 0$, but it has a degenerate minima on the surface such that $\sum_i (\gamma_i)^2 = v$, where $v = \sqrt{-m^2/\lambda}$. Now we pick a point on this loss minima and expand around it. Write $\gamma^i = (\pi^k, v + \sigma)$, where $k \in \{1, \ldots, D' - 1\}$. Intuitively, the $\pi^k$ fields are in the subspace of degenerate minima and $\sigma$ is the field orthogonal to $\pi$. Then it can be shown that the Lagrangian can be written as

$$\mathcal{L} = \mathcal{T}_\pi + \mathcal{T}_\sigma - \mathcal{V}_\pi - \mathcal{V}_\sigma - \mathcal{V}_{\pi\sigma},$$

where, in the weak coupling limit $\lambda \to 0$,

$$
\begin{aligned}
\mathcal{T}_\pi &= \frac{1}{2}(\partial_t \pi^k)^2 - \frac{1}{2}(\partial_\mathbf{z} \pi^k)^2, \\
\mathcal{T}_\sigma &= \frac{1}{2}(\partial_t \sigma)^2 - \frac{1}{2}(\partial_\mathbf{z} \sigma)^2, \\
\mathcal{V}_\pi &= O(\lambda), \\
\mathcal{V}_\sigma &= -\frac{1}{2}m^2\sigma^2, \\
\mathcal{V}_{\pi\sigma} &= O(\lambda),
\end{aligned}
\tag{6}
$$

the fields $\pi$ and $\sigma$ decouple from each other and can be treated separately. The $\sigma$ fields satisfy the Klein-Gordon Equation $(\Box - m^2)\sigma = 0$, with $\Box = \partial_t^2 - \partial_{\mathbf{z}}^2$. The $\pi$ fields satisfy the wave-equation, $\Box\pi = 0$. The correlation functions of the weights across sample space and layers, $P_\sigma = \langle\sigma(\mathbf{z}',t')\sigma(\mathbf{z},t)\rangle$ and $P_\pi = \langle\pi(\mathbf{z}',t')\pi(\mathbf{z},t)\rangle$ are the Green's functions of the respective equations of motion. Fourier transforming the correlation functions give

$$P_{\sigma,\pi}(t,\mathbf{k}) = \frac{i}{2\omega_0}\exp\left(-i\omega_0 t\right),\tag{7}$$

where $\omega_0 = \sqrt{|\mathbf{k}|^2 + |m_{\sigma,\pi}^2|}$, and $m_\pi^2 = \frac{1}{4}\lambda\eta^2 \simeq 0$. The correlation function $P_\pi$ is dominated by values of $|\mathbf{k}| \simeq 0$. Therefore $\langle\pi\pi\rangle \to \infty$ as $\lambda\eta^2 \to 0$. On the other hand, it can be shown that $\langle\sigma\sigma\rangle$ is damped by the weight eigenvalues $|m^2|$. The singularity in the correlation function means that the value of the weights at the start of the layer is highly correlated with the ones in later layers.

In the language of group theory. The $O(D)$ symmetry is broken down to $O(D-1)$. Elements of $O(D)$ are the $D \times D$ orthogonal matrices, which have $D(D-1)/2$ independent continous symmetries (e.g. the Euler angles in $D = 3$). The number of continuous broken from $O(D)$ to $O(D-1)$ is $D-1$. In the above example we showed that this corresponds to the $D-1$ $\pi^k$ fields. Each of which have zero Hessian eigenvalue.

Even though we formulated our field theory based on the decoupling limit of ResNets, the result of infinite correlation is very general and can be applied even if the decoupling limit is not valid. It is a direct result of spontaneous symmetry breaking. We state the Goldstone Theorem without proof.

**Theorem (Goldstone):** For every continuous symmetry that is spontaneously broken, a weight $\pi$ with zero Hessian eigenvalue is generated at zero temperature (learning rate $\eta$). $\Box$

## D   WHY QUANTUM FIELD THEORY?

In brief, the formalism for spontaneous symmetry breaking is mostly done in quantum field theory. In terms of statistics, quantum mechanics is the study of errors. We also believe that it is a good approximation to deep neural networks in the presence of the non-linear operators. The non-linear operators quantizes the input. Let $R$ denotes the opertor corresponding to a sigmoid, say, then the output is $R(\mathbf{W}) \simeq \{0, +1\}$ for the most part. And the negative end of ReLU is zero.

Let us take a step back and go through the logical steps to understand that a scalar quantum field theory *is* perhaps one of the simplest model one can consider to describe a neural network layer by layer, in the decoupling limit. We wish to formulate a dynamical model to describe the weights layer by layer,

1. We know that the outputs of non-linearities are quantized. And they need to be quantized to break the affine symmetry (see Theorem 1). This leads to quantum mechanics.
2. Quantum mechanics does not admit spontaneous symmetry breaking (Zee, 2010).
3. The decoupling limit allows spontaneous symmetry breaking in quantum mechanics.
4. The decoupling limit and non-linearity together is quantum field theory.

Therefore, if one wishes to model the outputs of non-linearities in the decoupling limit. There is no choice but to employ quantum field theory.

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
