# OpenReview forum: "Spontaneous Symmetry Breaking in Deep Neural Networks"
_ICLR.cc/2018/Conference — Reject_

### Official Review · AnonReviewer2 · 2017-11-26
**Do we really need quantum field theory?**

**Rating:** 3
**Confidence:** 4

**Review:**

In this paper, an number of very strong (even extraordinary) claims are made:

* The abstract promises "a framework to understand the unprecedented performance and robustness of deep neural networks using field theory."
* Page 8 states that this is "This is a first attempt to describe a neural network with a scalar quantum field theory."
* Page 2 promises the use of the "Goldstone theorem" (no less) to understand phase transition in deep learning
* It also claim that many "seemingly different experimental results can be explained by the presence of these zero eigenvalue weights."
* Three important results are stated as "theorem", with a statement like "Deep feedforward networks learn by breaking symmetries" proven in 5 lines, with no formal mathematics.

These are extraordinary claims, but  when reaching page 5, one sees that the basis of these claims seems to be the Lagrangian of a simple phi-4 theory, and Fig. 1 shows the standard behaviour of the so-called mexican hat in physics, the basis of the second-order transition. Given physicists have been working on neural network for more than three or four decades, I am surprise that this would enough to solve all these problems!

I tried to understand these many results, but I am afraid I cannot really understand or see them. In many case, the explanation seems to be a vague analogy. These are not without interest, and maybe there is indeed something deep in this paper, but it is so far hidden by the hype. Still, I fail to see how the fact that phase transitions and negative direction in the landscape is a new phenomena, and how it explains all the stated phenomenology. Beside, there are quite a lot of things known about the landscape of these problems

Maybe I am indeed missing something, but i clearly suspect the authors are simply overselling physics results.

I have been wrong many times, but I beleive that the authors should probably precise their claim, and clarify the relation between their results and both the physics AND statistics litterature, or better, with the theoretical physics litterature applied to learning, which is ---astonishing-- absent in the paper.

About the content:

The main problem for me is that the whole construction using field theory seems to be used to advocate for the appearence of a phase transition in neural nets and in learning. This rises three comments:

(1) So we really need to use quantum field theory for this? I do not see what should be quantum here (despite the very vague remarks page 12 "WHY QUANTUM FIELD THEORY?")


(2) This is not new. Phase transitions in learning in neural nets are being discussed since aboutn 40 years, see for instance all the pionnering work of Sompolinky et al. one can see for instance the nice review in https://arxiv.org/abs/1710.09553 In non aprticular order, phase transition and symmetry breaking are discussed in
* "Statistical mechanics of learning from examples", Phys. Rev. A 45, 6056 – Published 1 April 1992
* "The statistical mechanics of learning a rule", Rev. Mod. Phys. 65, 499 – Published 1 April 1993
* Phase transitions in the generalization behaviour of multilayer neural networks
http://iopscience.iop.org/article/10.1088/0305-4470/28/16/010/meta
* Note that some of these results are now rigourous, as shown in "Phase Transitions, Optimal Errors and Optimality of Message-Passing in Generalized Linear Models", https://arxiv.org/abs/1708.03395
* The landscape of these problems has been studied quite extensivly, see for instance "Identifying and attacking the saddle point problem in high-dimensional non-convex optimization", https://arxiv.org/abs/1406.2572


(3) There is nothing particular about deep neural net and neural nets about this. Negative direction in the Hessian in learning problems appears in matrix and tensor factorizaion, where phase transition are well understood (even rigorously, see for instance, https://arxiv.org/abs/1711.05424 ) or in problems such as unsupervised learning, as e.g.:
https://journals.aps.org/prl/abstract/10.1103/PhysRevLett.86.2174
https://journals.aps.org/pre/pdf/10.1103/PhysRevE.50.1766

Here are additional comments:

PAGE 1:

* "It has been discovered that the training process ceases when it goes through an information bottleneck (ShwartzZiv & Tishby, 2017)".

While this paper indeed make a nice suggestion, I would not call it a discovery yet as this has never been shown on a large network. Beside, another paper in the conference is claiming exacly the opposite, see : "On the Information Bottleneck Theory of Deep Learning". This is still subject of discussion.

* "In statistical terms, a quantum theory describes errors from the mean of random variables. "

Last time I studied quantum theory, it was a theory that aim to explain the physical behaviours at the molecular, atomic and sub-atomic levels, usinge either on the wave function (Schrodinger) or the Matrix operatir formalism (Hesienbger) (or if you want, the path integral formalism of Feynman).

It is certainly NOT a theory that describes errors from the mean of random variables. This is, i beleive, the field of "statistics" or "probability" for correlated variables. It is certianly used in physics, and heavily both in statistical physics and in quantum thoery, but this is not what the theory is about in the first place.

Beside, there is little quantum in this paper, I think most of what the authors say apply to a statistical field theory ( https://en.wikipedia.org/wiki/Statistical_field_theory )

* "In the limit of a continuous sample space, the quantum theory becomes a quantum field theory."

Again, what is quantum about all this? This true for a field theory, as well for continous theories of, say, mechanics, fracture, etc...

PAGE 2:

* "Using a scalar field theory we show that a phase transition must exist towards the end of training based on empirical results."

So it is a scalar classical field theory after all. This sounds a little bit less impressive that a quantum field theory. Note that the fact that phase transition arises in learning, and in a statistical theory applied to any learning process, is an old topic, with a classical litterature. The authors might be interested by the review "The statistical mechanics of learning a rule", Rev. Mod. Phys. 65, 499 – Published 1 April 1993

PAGE 8:


* "In this work we solved one of the most puzzling mysteries of deep learning by showing that deep neural networks undergo spontaneous symmetry breaking."

I am afraid I fail to see what is so mysterious about this nor what the authors showed about it. In any case, gradient descent break symmetry spontaneously in many systems, including phi-4, the Ising model or (in learning problems) the community detection problem (see eg https://journals.aps.org/prx/abstract/10.1103/PhysRevX.4.011047). I am afraid I miss what is new there...

* "This is a first attempt to describe a neural network with a scalar quantum field theory."

Given there seems to be little quantum in the paper, I fail to see the relevance of the statement. Secondly, I beleive that field theory has been used, many times and in greater lenght, both for statistical and dynamical problems in neural nets, see eg.
* http://iopscience.iop.org/article/10.1088/0305-4470/27/6/016/meta
* https://arxiv.org/pdf/q-bio/0701042.pdf
* http://www.lps.ens.fr/~derrida/PAPIERS/1987/gardner-zippelius-87.pdf
* http://iopscience.iop.org/article/10.1088/0305-4470/21/1/030/meta
* https://arxiv.org/pdf/cond-mat/9805073.pdf

---

### Official Review · AnonReviewer3 · 2017-11-26
**Hard to follow**

**Rating:** 3
**Confidence:** 3

**Review:**

The paper makes a mathematical analogy between deep neural networks and quantum field theory, and claims that this explains a large number of empirically observed phenomena.

I have a solid grasp of the relevant mathematics, and a superficial understanding of QFT, but I could not really make sense of this paper. The paper uses mathematics in a very loose manner. This is not always bad (an overly formal treatment can make a paper hard to read), but in this case it is not clear to me that the results are even "correct modulo technicalities" or have much to do with the reality of what goes on in deep nets.

The first thing I'm confused about is the nature and significance of the symmetries considered in this paper. At a very high level, there are two kinds of symmetries one could consider in DL: transformations of the input space that leave invariant the desired output, and transformations of the weight space that leave invariant the input/output mapping. These are not necessarily related. For instance, a translation or rotation of an image is an example of the former, whereas an arbitrary permutation of hidden units (and corresponding rows/columns of weight matrices) is an example of the latter. This paper is apparently dealing with groups that act on the input as well as the weight space, seemingly conflating the two.

Section 2.2 defines the action of symmetries on the input and weight space. For each layer t, we have a matrix Q_t in G, where G is an unspecified Lie group. Since all Q_t are elements of the same group, they have the same dimension, so all layers must have the same dimension as well. This is somewhat unrealistic. Furthermore, from the definitions in 2.2 it seems that in order to get covariance, the Q_t would have to be the same for all t, which is probably not what the authors had in mind.

For symmetries like rotation/translation of images, a better setup would probably involve a single group with different group actions or linear group representations for each layer. In that case, covariance of the weight layers is not automatic, but only holds for certain subspaces of weight space. For permutation or scale symmetries in weight space, a more sensible setup would be to say that each layer has a different group of symmetries, and the symmetry group of the whole network is the direct product of these groups.

It is stated that transformations in the affine group may not commute with nonlinearities, but rotations of feature maps do. This is correct (at least up to discretization errors), but the paper continues to talk about affine and orthogonal group symmetries. Later on an attempt is made to deal with this issue, by splitting the feature vectors into a part that is put to zero by a ReLU, and a part that is not, and the group is split accordingly. However, this does not make any sense because the pattern of zeros/non-zeros is different for each input, so one cannot speak of a "remnant symmetry" for a layer in general.

The connection between DL and QFT described in 2.3 is based on some kind of "continuous limit" of units and layers, i.e. having an uncountably infinite number of them. Even setting aside the enormous amount of technical difficulty involved in doing this math properly, I'm a bit skeptical that this has anything to do with real networks.

As an example of how loose the math is, "theorem 1" is only stated in natural language: "Deep feedforward networks learn by breaking symmetries". The proof involves assuming that the network is a sequence of affine transformations (no nonlinearities). Then it says that if we include a nonlinearity, it breaks the symmetry. Thus, since neural nets use nonlinearities, they break symmetries, and therefore learning works by breaking symmetries and the layers can learn a "more generalized representation" than an affine network could. The theorem is so vaguely stated that I don't know what it means, and the proof is inscrutable to me.

Theorem 2 states "Let x^T x be an invariant under Aff(D)". Clearly x^T x is not invariant under Aff(D).

The paper claims to explain many empirical facts, but it is not exactly clear which are the conspicuous and fundamental facts that need explaining. For instance, the IB phase transition claimed to happen in deep learning was recently called into question [1]. It appears that this phenomenon does not occur in ReLU nets but only in sigmoid nets, but the current paper purports to explain the phenomenon while assuming ReLUs. I would further note that the paper claims to explain a suspiciously large number of previously observed phenomena (Appendix A), but as far as I can tell does not make novel testable predictions.

The paper makes several strong claims, like "we [...] illustrate that spontaneous symmetry breaking of affine symmetries is the sufficient and necessary condition for a deep network to attain its unprecedented power", "This phenomenon has profound implications", "we have solved one of the most puzzling mysteries of deep learning", etc. In my opinion, unless it is completely obvious that this is indeed a breakthrough, one should refrain from making such statements.

[1] On the information bottleneck theory of deep learning. Anonymous ICLR2018 submission.

---

### Official Review · AnonReviewer1 · 2017-12-02
**Difficult to parse**

**Rating:** 3
**Confidence:** 3

**Review:**

The paper promises quite a few intriguing connections between information bottleneck, phase transitions and deep learning. While we think that this is a worthwhile bridge to build between machine learning and statistical field theory, the exposition of the paper leaves much to be desired. Had it been a clean straightforward application of QFT, as well-trained theoretical physicists, we would have been able to evaluate the paper.
Generally, it would help the reader to have an overall map and indication of the steps that would be taken formally.
Specifically, starting from Section 2.3, especially around the transition to continuous layers, very little information is provided how one is dealing with the cost function and the results are derived. Section 2.3 would benefit from expanded discussion with examples and detailed explanations.
Minor:
The following sentence in third paragraph of the Introduction is incomplete:
Because the ResNet does
not contain such symmetry breaking layers in the architecture.

---

### Decision · Program_Chairs · 2018-01-29
**ICLR 2018 Conference Acceptance Decision**

**Decision:**

Reject

**Comment:**

The paper makes overly strong claims, too weakly supported by a hard to follow and insufficiently rigorous mathematical argument. Connections with a large body of relevant prior literature are missing.